# *Pseudomonas taetrolens* ULE-PH5 and *Pseudomonas* sp. ULE-PH6 Isolated from the Hop Rhizosphere Increase Phosphate Assimilation by the Plant

**DOI:** 10.3390/plants13030402

**Published:** 2024-01-29

**Authors:** Seyedehtannaz Ghoreshizadeh, Carla Calvo-Peña, Marina Ruiz-Muñoz, Rebeca Otero-Suárez, Juan José R. Coque, Rebeca Cobos

**Affiliations:** Instituto de Investigación de la Viña y el Vino, Escuela de Ingeniería Agraria, Universidad de León, 24009 León, Spain; sghore00@estudiantes.unileon.es (S.G.); ccalp@unileon.es (C.C.-P.); mruim@unileon.es (M.R.-M.); rebeca.osuarez@unileon.es (R.O.-S.)

**Keywords:** phosphate-solubilizing bacteria, hop, rhizosphere, plant growth-promoting rhizobacteria, perennial crops

## Abstract

Most of the phosphorus incorporated into agricultural soils through the use of fertilizers precipitates in the form of insoluble salts that are incapable of being used by plants. This insoluble phosphorus present in large quantities in soil forms the well-known “phosphorus legacy”. The solubilization of this “phosphorus legacy” has become a goal of great agronomic importance, and the use of phosphate-solubilizing bacteria would be a useful tool for this purpose. In this work, we have isolated and characterized phosphate-solubilizing bacteria from the rhizosphere of hop plants. Two particular strains, *Pseudomonas taetrolens* ULE-PH5 and *Pseudomonas* sp. ULE-PH6, were selected as plant growth-promoting rhizobacteria due to their high phosphate solubilization capability in both plate and liquid culture assays and other interesting traits, including auxin and siderophore production, phytate degradation, and acidic and alkaline phosphatase production. These strains were able to significantly increase phosphate uptake and accumulation of phosphorus in the aerial part (stems, petioles, and leaves) of hop plants, as determined by greenhouse trials. These strains are promising candidates to produce biofertilizers specifically to increase phosphate adsorption by hop plants.

## 1. Introduction

The productivity of agricultural soils around the world largely depends on the use of both organic and inorganic fertilizers to increase crop productivity. The use of these fertilizers aims to provide the nutrients required by the plant. Among these nutrients, phosphorus (P) plays an important role since it is involved in crucial biochemical processes including the biosynthesis of nucleic acids, lipid metabolism, especially the biosynthesis of phospholipids, which are essential components of cellular membranes, energy metabolism, and photosynthesis [1]. Its cycle in the biosphere can be described as “sedimentary”, because there is no interchange with the atmosphere, and unlike the case for nitrogen, no large atmospheric source can be made biologically available [2,3]. This fact determines that, with the current methods of production of P-based inorganic fertilizers, P can be considered a limited, non-renewable natural resource [4,5]. Indeed, P-enriched fertilizers are obtained from P extracted by mining techniques from phosphate rock. The main reserves of phosphate rock are located in China, Morocco, Russia, the USA, and South Africa [6]. Some studies suggest that at the current rate of mineral extraction, phosphate rock reserves could be depleted over the next two centuries, with more alarmist ones claiming it could even happen in the next 50 years [7].

Phosphorus is one of the most limited nutrients in global agriculture [8], particularly in tropical areas [9], due to its low availability in the soil. On average, the P content in soils is around 0.05% [2,3], although most of it (95–99%) is not available for plants due to its presence in the form of highly insoluble phosphates and fixing on organic matter [10]. In fact, less than 1% of total P in soils is present in the soil solution and is considered available to plants, and accordingly, P is a major limiting factor for plant growth and crop productivity [4,5]. In an attempt to reverse this situation, farmers all around the world fertilize their plots by adding large amounts of P-based mineral fertilizers or organic fertilizers rich in P to satisfy crop nutritional requirements. However, most of this P is almost immediately fixed by soils due to the precipitation of insoluble salts or linking to organic matter, leading to the accumulation in soils of a “legacy P” [5,11]. For all these reasons, this situation is unsustainable in the long term, especially if we take into consideration that the accumulation of “legacy P” in soils and potential transfer to water bodies ultimately leads to environmental concerns of eutrophication [12]. In this context, the use of phosphorus-solubilizing bacteria (PSB) could represent a promising management strategy for improving P-use efficiency. PSBs could also be an aid in the mobilization of this “legacy P”, which, according to some studies, would be enough to sustain maximum crop yields worldwide for about 100 years if they were mobilized [13,14].

PSBs have been isolated from a wide variety of environments, including agricultural and rhizosphere soils and different kinds of sludges. Most of the more efficient PSBs isolated belong to the *Pseudomonas* and *Bacillus* genera [4,11,15,16,17,18,19,20,21]. However, efficient phosphate solubilizing capability has also been reported for bacterial strains belonging to other genera such as *Advenella* and *Burkholderia* [21], *Azotobacter* and *Azospirillum* [22], *Thiobacillus* [4], *Rhizobium* [2,23], and *Streptomyces* [15,24].

Many of these strains have been tested on different annual crops, including barley [21,24], canola [22], maize [11,25], peanut [22,26], potato [22], sesame [27], sorghum [22], soybean [28], wheat [23], and wild mint [29], to cite some of the most representative examples.

PSBs have the ability to use different molecular mechanisms for inorganic phosphate solubilization and mobilization [30,31]. Among these mechanisms, acidolysis, which involves the secretion of organic (a common trait), or inorganic acids (a rarer trait), is one of the most important. In fact, the secretion of organic acids like acetic, butyric, fumaric, glutamic, 2-ketogluconic, lactic, malic, malonic, oxalic, propionic, saccharinic, succinic, and tartaric acids during microbial growth is well documented [21,32,33]. Organic phosphate-containing compounds can account for up to 30–50% of the total amount of P present in soils, primarily in the form of inositol phosphate (soil phytate). Other organic phosphorus compounds present in soils include phosphomonoesters, phosphodiesters, phospholipids, nucleic acids, and phosphotriesters. Additionally, large quantities of xenobiotics, such as pesticides, detergents, antibiotics, and flame retardants, that are regularly released into the environment are also known to contain organic P [30]. Phosphorus present in phytates can be mobilized by specialized phytase enzymes [34]. On the other hand, mineralization of P present in other organophosphorus compounds is mainly carried out by dephosphorilation, which is catalyzed by different enzymes, including phosphatases (both acidic and alkaline) and C-P cleaving enzymes [4,31,35]. Chelation and complexation are two other important mechanisms of P solubilization, which are mainly mediated by the production and secretion of siderophores [35,36] and extracellular polysaccharides [37]. Siderophores are small molecules with low molecular weights, and under low iron stress, phosphate-dissolving bacteria produce siderophores that chelate Fe^3+^, Al^3+^, and Ca^2+^ in the soil, releasing phosphate ions for phosphate dissolution [31,38]. Finally, soil P can be mobilized by reductive dissolution of Fe(Al)-complexed organic P [38,39].

This work is focused on the isolation and characterization of phosphate-solubilizing bacteria (PSBs) isolated from the rhizosphere environment of hop plants. The best solubilizing isolates were characterized by analyzing other interesting traits for plant growth promotion and tested in pot trials in a greenhouse to analyze their capability to enhance P assimilation by the plant. The final objective of this work is to demonstrate that PSBs can also be beneficial for perennial crops by stimulating P uptake and assimilation by the plant or producing any change in nutritional properties after treatment.

## 2. Results

### 2.1. Isolation of Phosphate-Solubilizing Bacteria on Solid Media and Their Identification

A total of 634 bacterial isolates belonging to 63 morphologically distinct bacteria were isolated from hop plant rhizosphere soil. These isolates were identified based on easily observable macroscopic (size, color, appearance, colony shape, and pigment production) and microscopic (cellular morphology, spore production, and mobility) traits. When these isolates were replicated onto NBRIP solid agar plates, 113 isolates (17.82% of the tested isolates) were able to produce clear halozones, indicating phosphate solubilization. The Solubilization Index (SI) of the rhizosphere isolates ranged from 1.67 to 8.31, as estimated at 7 days after inoculation. The capability to degrade phytate was also tested by replicating all the isolates in phytase screening medium (PSM). A total of 87 isolates (13.72%) were able to degrade phytate with a SI (estimated at 5 days of growth) ranging between 1.87 and 8.31%. Five strains exhibiting a SI value higher than 5.0 in both agar media were selected for molecular identification (Table 1).

The sequencing of the 16s rRNA confirmed that four out of the five isolates (ULE-PH1, ULE-PH5, ULE-PH6, and ULE-PH12) belonged to the *Pseudomonas* sp. genus. However, only the ULE-PH1 and ULE-PH5 isolates could be identified at the species level. Indeed, isolate ULE-PH1 was identified as *P. cedrina* since the 16S rRNA sequence obtained showed 100% homology with several isolates belonging to this species, whereas the *rpoD* partial sequence showed a 99.35% homology to the corresponding gene of the same species (Table 1). This value was higher than 98.0% defined as the cutoff value to differentiate *Pseudomonas* sp. strains at the species level by using this molecular marker, according to [40]. Partial sequencing of the 16S rRNA gene from isolate ULE-PH5 showed inconclusive results since nucleotide homologies higher than 99.60% were obtained with several *Pseudomonas* species, including *P. versuta*, *P. taetrolens*, and *P. fragi*. However, partial sequencing of the *rpoD* marker allowed its identification as *P. taetrolens* (99.81%) (Table 1).

Isolate ULE-PH6 could not be assigned to a particular species of the genus *Pseudomonas* since nucleotide homologies higher than 99.8% were obtained with several species and *rpoD* nucleotide homologies higher than 98.0% were obtained with both *P. alvandae* and *P. canavaninivorans* species (Table 1), being this last a new species recently isolated from bean rhizosphere [41]. A similar situation was raised for the ULE-PH12 isolate since nucleotide homology higher than 99.90% for the 16S rRNA gene was detected. Sequencing of marker *rpoD* did not allow clear discrimination since homology higher than 98.00% was exhibited with *P. atacamensis* (99.02%) and *P. fluorescens* (98.53%).

Finally, isolate ULE-PH10 was identified as *Bacillus nitratireducens* since, although nucleotide homologies higher than 99.90% were detected with several *Bacillus* species, the partial sequence of the *rpoB* gene [42] showed a 99.46% homology with that particular species (Table 1).

### 2.2. Phosphate Solubilization in Liquid Medium

The ability to solubilize phosphate in NBRIP liquid medium was checked for all the strains listed in Table 1 up to 7 days of growth (Figure 1).

Under the experimental conditions tested, *Pseudomonas* sp. ULE-PH6 exhibited the highest solubilization capacity, reaching a maximum value of 1100.45 μM of soluble orthophosphate in the culture supernatant at 144 h (6 d) of growth. The remaining strains demonstrated a similar maximum solubilization capability, ranging between 472.83 μM for *P. taetrolens* ULE-PH5 (54 h), 467.51 μM for *P. cedrina* ULE-PH1 (72 h), 339.86 μM for *B. nitratireducens* ULE-PH10 (120 h), and 318.00 μM for *Pseudomonas* sp. ULE-PH12 (Figure 1) at 144 h of growth. At the end of the experiment, the pH of the culture supernatant was measured and determined to be 4.7 (ULE-PH1 strain), 4.5 (ULE-PH5), 4.3 (ULE-PH6), 4.4 (ULE-PH10), and 4.3 (ULE-PH12). These data indicated that all the strains were able to acidify the culture broth from the initial pH value of 7.0 to the indicated pH values.

Further studies selected *Pseudomonas* sp. ULE-PH6 and *P. taetrolens* ULE-PH5 due to their higher phosphate solubilization capability in liquid media, combined with higher values of SI in PSM medium (Table 1), indicating a high capability to degrade phytate in solid media.

### 2.3. Biochemical and Physiological Characterization of Isolated ULE-PH5 and ULE-PH6 Strains by Using API Tests

To achieve a deeper biochemical and physiological characterization of strains ULE-PH5 and ULE-PH6, an API^®^ 20 NE test was conducted. As shown in Table 2, both strains tested positive for the citochrome oxidase test, which is typical for strictly aerobic *Pseudomonas* sp. strains. However, both strains were unable to ferment glucose, although they could assimilate this sugar. They were also capable of assimilating arabinose, mannose, mannitol, gluconate, and capric, malic, and citric acids. The main differences between the two strains are that ULE-PH6 can reduce nitrate to nitrite and can assimilate N-acetyl glucosamine, while these traits are absent in strain ULE-PH5.

We also carried out an API^®^ 50 CH test that confirmed that both strains were able to efficiently oxidize D-glucose and D-mannose, whereas strain ULE-PH5 was also able to oxidize D-galactose (positive result at 24 h of incubation). ULE-PH5 was also able to oxidize slowly (positive result at 48 h of incubation) D-fructose, L-rhamnose, D-cellobiose, D-maltose, and D-melobiose. ULE-PH6 was able to slowly oxidize D-arabitol. Both strains were also able to slowly oxidize D-fucose.

### 2.4. Characterization of Miscellaneous Traits of Isolates ULE-PH5 and ULE-PH6

A study was conducted to test various traits of interest associated with PGPR to stimulate plant growth (Table 3). Both strains exhibited clear siderophore production in solid media. However, hydrogen cyanide (HCN) was only produced by the ULE-PH6 strain. Additionally, both strains were able to produce 3-indoleacetic acid (IAA), although strain ULE-PH5 exhibited a synthesis capacity of 59.29 μg/mL, practically double that detected in the ULE-PH6 strain (30.69 μg/mL). The ability to solubilize Zn was not detected in either of the two strains, whereas both strains were able to solubilize K (Table 3).

Although both strains were selected for their ability to solubilize inorganic phosphate in liquid medium, we also tested their ability to solubilize P associated with organic matter. Thus, we tested the phytase activity involved in the degradation of phytate, which is the main P storage compound in plants and is especially abundant in seeds [43]. Both strains exhibited a strong solubilization capability of phytate in solid media (Table 1). Accordingly, high activities of phytate degradation in liquid medium of 40.25 ± 1.71 mU/mL for ULE-PH5 and 43.38 ± 2.75 for ULE-PH6 were detected (Table 3).

The antifungal activity of isolates ULE-PH5 and ULE-PH6 against 2 of the most important fungal phytopathogens affecting hops, *Fusarium sambucinum* and *Verticillium dahliae*, was also tested. Unfortunately, the ULE-PH5 and ULE-PH6 strains exhibited very low antifungal activity against *Fusarium sambucinum* and *Verticillium dahliae*. The inhibition indexes of ULE-PH5 against *F. sambucinum* and *V. dahliae* were 34.2% and 25.3%, respectively. Similarly, the inhibition index of ULE-PH6 against *F. sambucinum* and *V. dahliae* was 36.6% and 24.0%, respectively.

Acid- and alkaline-phosphatases are important for the mineralization of phosphorus associated with organic matter [35]. Both activities were detected in the ULE-PH5 and ULE-PH6 strains. The levels of extracellular acidic phosphatases were considerably higher than the detected levels of extracellular alkaline phosphatases (Table 3). Acidic and alkaline phosphatases were also detected intracellularly, although their levels were, in all cases, lower than those detected outside the cell. In this case, levels of intracellular alkaline phosphatase were higher than those observed for acidic ones (Table 3).

### 2.5. Greenhouse Pot Experiments

A pot experiment was carried out to test the capability of ULE-PH5 and ULE-PH6 strains to favour P assimilation by hop plants. Plants inoculated with both ULE-PH5 and ULE-PH6 strains were able to accumulate a significantly higher amount of P (6835 ± 1106.17 ppm and 8015 ± 979.02 ppm, respectively) in the aerial parts of the plants (stems, petioles, and leaves) than negative control plants (5718 ± 124.36 ppm) (Figure 2). The assimilation level of P in plants inoculated with ULE-PH6 was quite similar to the values observed in positive control plants (7955 ± 979.56 ppm), which had been irrigated with a solution containing soluble phosphate, whereas the assimilation level in plants inoculated with ULE-PH5 was slightly lower (Figure 2).

The accumulation of phosphate in the root system was not significantly different in negative control plants (4617.72 ± 115.70) and those inoculated with ULE-PH5 (4836.55 ± 282.79) and ULE-PH6 (4344 ± 428.56) strains. On the contrary, the root system of positive control plants contained significantly higher levels of phosphate (5225.14 ± 218.74) when compared with plants from other treatments (Figure 2).

When the growth rate was analyzed by testing the dry weight of aerial parts (a mix of stems, petioles, and leaves) and roots, no significant differences were detected (Figure 3).

## 3. Discussion

Sustainable agriculture is a farming practice that meets society’s present textile and food needs in a sustainable manner without compromising the ability of current or future generations to meet their needs. It is associated with several key principles, including the use of decreased amounts of non-renewable and unsustainable inputs [44]. In this context, phosphorus (P) is demanding increasing attention given that the only known source of P used in fertilizer production is a finite resource: phosphate rock. At the current rate of phosphate rock extraction, its reserves could be depleted over the next two centuries, with more alarmist estimates suggesting depletion within the next 50 years [7]. As previously indicated in Section 1, much of the P added to soils in high amounts during many years of intensive fertilization is rapidly sequestered into insoluble and not available to plants P. This fertilization strategy is known as low-P use efficiency (PUE), and it has finally resulted in the accumulation in agricultural soils all over the world of a “legacy P” [45]. After this accumulation phase, farmers in many industrialized countries have been able to increase their PUE, often by using the accumulated residual soil P reserves [45,46,47]. In this scenario, the use of biofertilizers based on phosphate-solubilizing bacteria (PSBs) is an interesting tool to take advantage of this “legacy P”.

Numerous studies have reported that the use of phosphate-solubilizing bacteria (PSBs) can increase the productivity of many different annual crops, such as barley, canola, maize, peanut, potato, sesame, sorghum, soybean, and wheat, among others [11,21,22,24,25,26,27,28,29], as previously indicated in Section 1. However, to our knowledge, no data regarding the use of PSBs in perennial crops has been reported. Hops are a dioecious perennial climber crop that every spring develops from a system of rhizomes that remains buried under the ground. Rhizomes can be active for long time periods; their average life span is between 12 and 15 years, although there are references to plantations that have exceeded 25 years, with a production level maintained over time [48].

The isolation of PSBs from the rhizosphere environment of hop plants indicated that 17.82% of the isolates could solubilize phosphate in solid NBRIP medium. This value was, however, considerably lower than the value of 61.5% of PSBs isolated from the rhizosphere of barley plants [21], which might suggest that the capability of different vegetable species to recruit PSBs to their rhizosphere could be different. The characterization of the 5 best phosphate solubilizers in solid NBRIP medium indicated that 4 of them (isolates ULE-PH1, ULE-PH5, ULE-PH6, and ULE-PH12) belonged to the *Pseudomonas* genus, whereas 1 belonged to the *Bacillus* genus (isolate ULE-PH10). *Pseudomonas* species were also the most frequently isolated PSBs from the rhizosphere of barley plants (9 out of the 17; 52.94%) [21], and the rhizosphere of shisham plants (7 out of 18 isolates; 38.89%) [49]. They were also dominant [50]. PSB species were isolated from Tunisian soils [50]. Other studies reviewed by Zhu et al. [5] have also indicated that *Pseudomonas* is one of the most frequently isolated PSBs associated with different crops. Therefore, we can conclude that *Pseudomonas* seems to play a crucial, or at least an important, role from a quantitative point of view in the phosphate solubilization in the rhizosphere of many different vegetal species.

After analyzing phosphate solubilization in liquid media, a clear acidification of the culture supernatant was observed. At the end of the experiment, the pH values of the NBRIP medium ranged from 4.3 (ULE-PH6 and ULE-PH12 strains) to 4.7 (ULE-PH1), which were significantly lower than the initial pH of 7.0. The production of organic acids is a well-known mechanism used by many bacterial strains to solubilize phosphate [21,30,40,41]. These organic acids are produced in the periplasmic space by direct oxidation [51]. The release of these organic acids into the environment is accompanied by a decrease in pH [30]. Under low pH conditions, these organic acids can chelate metal ions in the soil (Fe^3+^, Al^3+^, and Ca^2+^) via hydroxyl and carbonyl groups. Their Ca^2+^ chelation ability is most significant, and they compete with phosphates for P adsorption sites in the soil, enhancing the soil uptake of phosphate and increasing the solubilizing capacity of inorganic P, resulting in increased solubility and availability of mineral phosphates [52]. In fact, it has been reported that different *Pseudomonas* species isolated from the rhizosphere of barley plants were able to produce and excrete from the culture supernatant different organic acids like citric, formic, and tartaric acids [21]. Although the production of different organic acids was not analyzed in our case, the data obtained from API^®^ 50 CH tests carried out for the ULE-PH5 and ULE-PH6 strains indicated that both strains are able to efficiently oxidize D-glucose and D-mannose, whereas strain ULE-PH5 was also able to oxidize D-galactose to produce, respectively, D-gluconic acid, D-mannaric acid, and D-Galactaric Acid. ULE-PH5 was also able to slowly oxidize D-fructose, L-rhamnose, D-cellobiose, D-maltose, and D-melobiose, whereas ULE-PH6 was able to slowly oxidize D-arabitol. Therefore, they could potentially produce the respective corresponding organic acids that could contribute to P solubilization.

The selection of PSBs is frequently carried out by taking into account other parameters or traits that are considered beneficial to promote plant growth and health. For instance, ULE-PH5 and ULE-PH6 isolates were found to produce Indole-3-Acetic Acid (IAA), which is considered one of the most important phytohormones [36,53]. IAA is an auxin that is related to the development of roots and leaves, such as lateral root elongation and leaf growth [1]. IAA production by rhizobacteria could also help activate the non-native plant response to resist biotic and abiotic stress conditions [1]. Therefore, it is expected that those strains that produce higher levels of IAA will enhance crop development [54,55] However, in the plant assays carried out, the plants inoculated with ULE-PH5 and ULE-PH6 strains did not show a higher root development than those plants of the negative and positive control. This data suggests that IAA is not produced in the experimental conditions tested, which, on the other hand, are quite restrictive for the development of the plant as it is forced to grow in a completely inorganic medium lacking soluble phosphate.

Hydrogen cyanide (HCN), which is produced by the isolate ULE-PH6, is another acid compound that could contribute to phosphate solubilization. HCN has been commonly described as a biocontrol factor against phytopathogens. However, it has been reported that it does not intervene in the toxicity against microorganisms per se, but HCN sequesters the iron present in the soil, as occurs with siderophores. Phytopathogen bio-control by iron competition has been described in siderophore producers, which are able to inhibit phytopathogen infections [56]. Moreover, HCN production could be an indirect mechanism of phosphorus solubilization in soil, in the same way that siderophore production is: iron sequestration could prevent it from binding free phosphate, which remains available in solution for its assimilation by plants [57,58].

Other types of compounds involved in the solubilization of inorganic phosphate are siderophores. Siderophores are low-molecular-weight organic ligands with high affinity and specificity for binding trivalent ions such as iron and aluminum [59]. Although their most prevalent role could be the capture of Fe^2+^ present in soils (iron is an essential nutrient for all bacteria), indirectly, siderophores decrease the iron concentration, which is able to react with the phosphorus, allowing its solubilization in this way [35,60]. Isolates ULE-PH5 and ULE-PH6 exhibited a good capability to produce siderophores [4].

Although most agricultural soils contain significant amounts of inorganic, insoluble phosphates, a significant proportion of the P unavailable to plants is fixed to the organic matter of soils or present in organic plant matter in the form of phytates [4]. Accordingly, microorganisms inhabiting bulk soils and rhizospheres have evolved to develop different mechanisms to extract and solubilize this P fixed in the organic fraction of soils. Phytate (inositol hexaphosphate) represents the major storage form of P and inositol in plants [61]. Although present in all vegetable tissues, it is especially abundant in pollen and seeds [62]. Many microorganisms inhabiting soils are able to efficiently degrade phytates by producing phytase enzymes. Since phytic acid ester bonds are highly stable and their natural degradation is practically impossible [63], the microbial mineralization of phytate by phytase plays an essential role in the process of P recycling [30]. Isolates ULE-PH5 and ULE-PH6 were both selected for their high phytase activity in solid medium, which was confirmed in liquid cultures.

Another mechanism used by soil microorganisms to recover P fixed to the organic material is the production of phosphatase enzymes [4], which have been widely documented among PBS [30,31]. Phosphatases induce the release of phosphorus from precipitated organic compounds in soil or fertilizers by dephosphorylating phosphoester or phosphoanhydride bonds. This hydrolysis removes a phosphate ion, generating a molecule with a free hydroxyl group and soluble phosphate [4,5,35]. Both ULE-PH5 and ULE-PH6 were able to produce extracellular alkaline and acidic phosphatases that allow the recovery of P in environmental conditions with a wide range of pHs. Phosphatases can be classified into three types: acidic, alkaline, and neutral phosphatases. Unlike acid and alkaline phosphatases, neutral phosphatases do not have a very pronounced effect on phosphorus mineralization and hydrolysis. Acidic phosphatases are more effective in mineralizing organic P in acidic soils with pH values less than 7, whereas alkaline phosphatases mainly catalyze the hydrolysis of phospholipids (i.e., phosphoglucose-6 and ATP) and release inorganic phosphorus in soils with pH values higher than 7 [31].

Finally, it is worth emphasizing that PSB could be of potential interest to enhance P assimilation and uptake by perennial crops. Therefore, it should not be surprising that in the coming years, the agricultural use of PSB will increase, as it could be applied to different perennial crops as a viable alternative and cost-effective way of stimulating the bioavailability of the P accumulated in soils as “legacy P”. This approach could significantly reduce the costs of fertilization based on fertilizers containing phosphates.

## 4. Materials and Methods

### 4.1. Isolation of Bacteria from the Rhizosphere of Hop Plants

Bacteria were specifically isolated from the rhizosphere of hop plants (Nugget cultivar) that had been cultivated according to conventional agronomical practices in a field located in León (Spain) (5°35′26.45″ W and 42°35′2.524″ N). Samples were collected from 5 different plants after digging to access the root system of the plant. After leaving the root system, exposed rhizosphere soil in direct physical contact with the roots was collected by using a sterile spatula. Rhizosphere soil samples were transferred to sterile Falcon tubes (50 mL), kept in an icebox, and preserved 117 at 4 °C until processing. Later, 1 g of soil was weighed, and tenfold serial dilutions were made in saline solution (NaCl 0.9% *w*/*v*), and 0.1 mL aliquots of each dilution were spread onto Nutrient Agar (Condalab, Torrejón de Ardoz, Spain) and Plate Count Agar (Condalab) supplemented with natamycin (200 μg/mL) to avoid fungal growth. Plates were incubated at 30 °C for up to 8 days. Bacteria representative of different macro (color, colony shape, colony border, consistency, etc.) and microscopic morphological types (as determined after staining with methylene blue or Gram stain) were selected at random and conserved in Nutrient Agar plates at 4 °C until use.

### 4.2. Analysis of Phosphate Solubilization and Phytase Production in Solid Media

All the strains were checked for phosphate solubilization on solid media by streaking onto NBRIP (National Botanical Research Institute’s Phosphate) agar [64]. Colonies surrounded by clarification halos (halozones) were selected to analyze their solubilization index (SI) according to the protocol described by Mardad et al. [65]. Briefly, positive strains were grown on NBRIP solid medium plates that were incubated at 30 °C for up to 7 days. SI was determined by measuring, at 7 days of incubation, the halo and the colony diameter according to the next formula: SI = (colony diameter + halozone diameter)/(colony diameter). Solubilization assays were carried out in triplicate. Isolates were also tested for extracellular phytase production in Phytase Screening Medium (PSM), which contained per liter: D-glucose, 2.0 g; sodium phytate, 0.4 g; CaCl_2_, 0.2 g; NH_4_NO_3_, 0.5 g; KCl, 0.05 g; MgSO_4_ × 7H_2_O, 0.05 0 g; FeSO_4_ × 7H_2_O, 0.001 g; MnSO_4_ × 5H_2_O, 0.001 g; and agar, 2.0 g, adjusted to pH 6.0 [66]

### 4.3. Molecular Identification of Bacterial Isolates

Those five strains exhibiting a SI value higher than 5.0 in both NBRIP medium and Phytase screening medium were selected for molecular identification (Table 1). Genomic DNA was extracted as described by Hopwood et al. [67]. Amplification of 16S rRNA genes was carried out using primers 27F and 1492R [68]. For identification of *Pseudomonas* strains, amplification of the *rpoD* gene was performed according to the protocol described by Girard et al. [40] by using PsEG30F and PsEG790R primers. Identification of *Bacillus* species was carried out by partial sequencing of the *rpoB* gene using primers BA-RF and BA-RR and PCR conditions described by Ko et al. [42]. DNA sequences had been deposited in the GenBank^®^ under the Accession numbers listed in Table 4.

Isolates were identified by comparison with the corresponding sequences of type strains found in the Ez Taxon-e database [69] (http://www.ezbiocloud.net/eztaxon/identify) (accessed on 30 November 2023). Sequence alignment as well as phylogenetic trees were done using the MEGA 6 software [70].

### 4.4. Analysis of Phosphate Solubilization in Liquid Media

Those strains with the higher SI values were checked for phosphate solubilization in NBRIP liquid medium, using tri-calcium phosphate as the sole P source, as previously reported [21]. Briefly, Erlenmeyer flasks (500 mL) containing 100 mL of NBRIP liquid medium were inoculated at a final concentration of 5 × 10^4^ cfu/mL with each bacterial strain tested. Assays were carried out at 30 °C and 150 rpm for 20 days. Bacterial cells and insoluble phosphate were removed by centrifugation at 14,000 rpm for 10 min. The pH of the cultures was determined by a pH meter. Once the cultures were completed, samples of 100 µL of the diluted supernatant were mixed with 800 µL of 0.045% (*w*/*v*) malachite green and 100 µL of 34% (*w*/*v*) sodium citrate. Samples were incubated at 25 °C for 30 min in the dark. The absorbance of the mix samples was measured at 660 nm, and the quantity of soluble phosphorous was determined by comparison with a standard curve using K_2_HPO_4_. All experiments were carried out in triplicate.

### 4.5. API Tests

A more detailed biochemical and physiological characterization was carried out by using the API^®^ 20 NE system (Biomerieux, Marcy-l’Étoile, France) and API^®^ 50CH (Biomerieux) to analyze more specifically the carbohydrate metabolism. Both tests were carried out by following the procedures and methodologies indicated by the manufacturer.

### 4.6. Phosphatase Activity Determination

Phosphatase activities were determined as indicated by Magallon-Servin et al. [71]. Briefly, PSB was grown in 50-mL Erlenmeyer flasks containing 25 mL of TSB. Flasks were inoculated with 1 mL of an overnight-grown preculture, and they were incubated for 72 h at 28 °C on a rotary shaker (150 rpm). Bacterial cells were harvested by centrifugation, and culture supernatant was used to measure extracellular phosphatases. The cellular pellet was then washed twice in saline solution by centrifugation at 14,000 rpm for 20 min. Then cells were resuspended in 2 mL Tris-buffer (pH 7.0) and sonicated (VibraCell) on ice during 30 cycles of 2 s sonication and 15 s rest. The supernatant of sonicated cells containing intracellular phosphatases was recovered by centrifugation (14,000 rpm, 5 min at 4 °C) and used to determine enzymatic activities. Alkaline and acidic phosphatases were determined in 96-well microplates. Briefly, each plate-well received 50 μL of enzyme extract and 50 μL of substrate solution for acidic (0.05-M citrate buffer with 5.5 mM of nitrophenylphosphate at pH 4.8) or alkaline (0.05-M glycine buffer with 0.01% (*w*/*v*) MgCl_2_ × 6H_2_O and 5.5 mM of nitrophenylphosphate at pH 10.5) phosphatase. Microplates were incubated for 10 min at 37 °C on a rotary shaker (60 rpm). Reactions were stopped by adding 200 μL of stop solution (0.5 N NaOH), giving a total reaction volume of 300 μL. The amount of *p*-nitrophenol released by the phosphatases was quantified by reading the absorbance using a Microtiter Reader at 405 nm. As a standard, 300 μL of 0.05-μmol/mL *p*-nitrophenol were added to the plate. A blank (50 μL of buffer instead of enzyme) was run in parallel to account for any possible spontaneous hydrolyzation of 4-nitrophenyl phosphate during incubation. For determining specific activities, the total soluble protein in cell extracts and supernatants was measured by the Bradford method [72] adapted to microplates.

### 4.7. Qualitative and Quantitative Estimation of Phytase Production

Isolates were tested for extracellular phytase production in Phytase Screening Medium (PSM), which contained (*w*/*v*): 2.0 D-glucose; 0.4 sodium phytate; 0.2 CaCl_2_; 0.5 NH_4_NO_3_; 0.05 KCl; 0.05 MgSO_4_ × 7H_2_O; 0.001 FeSO_4_ × 7H_2_O; 0.001 MnSO_4_ × 5H_2_O; and 2.0 agar adjusted to pH 6.0 [66]. The isolates showing clear zones on PSM were further quantified in liquid medium following the methodology described by Qvirist et al. [73]. Briefly, the isolates were grown in LB liquid medium for 7 days at 30 °C, and cells were separated from the supernatant by centrifugation (8000 rpm at 4 °C for 5 min). Crude protein extract (75 µL) was incubated with 300 µL of phytate solution (1.5 mM of sodium phytate [phytic acid sodium salt hydrate] in 100 mM sodium acetate buffer, pH 5.0) for 30 min at 37 °C. The reaction was stopped by the addition of 750 µL of 5% trichloroacetic acid (*w*/*v*). A color reagent (750 µL), prepared freshly by mixing 4 volumes of 1.5% ammonium molybdate (*w*/*v*) in a 5.5% (*v*/*v*) sulphuric acid solution with 1 volume of 2.7% (*w*/*v*) ferrous sulfate solution, was added to the sample solution. The production of phosphomolybdate was measured spectrophotometrically at 700 nm. Standards were prepared from KH_2_PO_4_ in bi-distilled water at phosphate concentrations ranging from 0 to 20 mg/L.

### 4.8. Determination of Miscellaneous Activities Related to PGPR Traits

Selected PSB were checked for siderophore production on CAS agar medium containing chrome azurol (CAS) and hexadecyltrimethylammonium bromide (HDTMA) as indicators [74]. The bacterial cultures were inoculated into the CAS medium and incubated at 28 ± 2 °C for 5 days. The colony with a yellow-orange color halo zone was indicated as positive for siderophore production. In order to achieve quantitative estimation of indole acetic acid (IAA) production, cultures were grown at 30 °C and 150 rpm for 24 h. Flasks containing 100 mL of Nutrient Broth (NB) media supplemented with 1 g/L tryptophan were inoculated at 10^4^ cells/mL and incubated at 30 °C and 150 rpm for 10 days. 1.5 mL samples were taken every 24 h to determine growth spectrophotometrically and IAA production by using the Salkowski method [75]. Briefly, 1 mL of culture supernatants were mixed with 2 mL of Salkowski reagent and incubated in the dark for 25 min. DO was measured at 530 nm, and it was compared with a standard curve with indoleacetic acid ranging from 0 to 100 µg/mL. Potassium solubilization was detected using a modified Aleksandrov medium [76]. Zinc solubilization was tested according to plate assays on Tris-minimal medium supplemented with zinc oxide, according to Sharma et al. [77]. Solubilization indexes were estimated as the quotient between the diameter of the solubilization halo zone and the diameter of the bacterial colony, multiplied by 100. HCN production in agar plates was tested as reported by Khan et al. [78]. Briefly, nutrient broth was amended with 4.4 g glycine/L, and isolates were streaked on a modified agar plate. A Whatmann filter paper No. 1 filled with 2% sodium carbonate in a 0.5% picric acid solution was placed on top of the plate. Plates were sealed with parafilm and incubated at 28 ± 2 °C for 4 days. The development of orange to red color indicated HCN production: paper color from yellow to light brown, brown, or reddish brown was recorded for weak (+), moderate (++), and strong (+++) reactions, respectively. Putative antifungal activity against *Verticillium dahliae* and *Fusarium sambucinum* was tested by the dual culture technique as previously described in 9-cm Petri dishes [21].

### 4.9. Greenhouse Pot Experiments

Rhizomes of Cascade variety hop plants were obtained after routine pruning of the root system of 10-year-old plants carried out in spring 2023. Rhizomes of similar caliber and weight (20 ± 1 g) were selected.

Rhizomes were placed in 5 L pots containing vermiculite as substrate mixed with tricalcium phosphate at a 25:1 ratio. Before filling the pots, the vermiculite was washed with distilled water and sterilized twice by autoclaving at 121 °C for 20 min. Rhizomes were superficially disinfected by immersing in 70% ethanol for 1 min and 3% (*v*/*v*) sodium hypochlorite for 5 min, respectively. After that, the rhizomes were extensively washed with sterilized milli-Q water 3 times for 5 min. Next, rhizomes were inoculated by immersion in a culture containing 10^8^ colony-forming units (cfu)/mL of each individual plant growth-promoting rhizobacteria (PGPR) tested for 30 min. In each pot, a rhizome was buried in the vermiculite at a distance of 8–10 cm from the surface of the substrate. Negative and positive controls were not inoculated. A 3 m high bamboo pole was inserted into the bottom of each pot to be used as a support to allow the plants to climb. The experiment was finished when 50% of the control plants (not inoculated) reached the top of the bamboo pole. Two hundred mL of modified plant nutrient solution [79], without any phosphate source, was supplied weekly to each pot. Whereas positive controls were irrigated with a plant nutrient solution supplemented with 0.2 g/L of soluble KH_2_PO_4_. For evaluation of the growth promotion effect of each selected PGPR, the dry weight of the roots and the aerial part of the plants (stems, petioles, and leaves) were measured. Dry weight was determined by placing the samples separately in an oven at 60 °C for 120 h. Assimilated P was determined at the root and aerial parts, as described by Watanabe and Olsen [80].

## 5. Conclusions

The main conclusions of this study are as follows: (1) *Pseudomonas* species are the predominant PSBs in the rhizosphere of hop plants; (2) The application of 2 particular strains, *P. taetrolens* ULE-PH5 and *Pseudomonas* sp. ULE-PH6, to rhizomes of hop plants was able to stimulate phosphate assimilation and accumulation in the aerial parts of the plant when they were growth in a substrate with insoluble P as the only source; (3) Both strains also exhibited other plant growth promoting traits including phytate degradation and acidic and alkaline phohphatases production involved in the solubilization of P linked to organic matter; (4) Finally, this work demonstrates that PSB can also be effective to promote P assimilation by perennial crops and therefore their putative use is a promising strategy for capitalizing on the currently unused legacy P present in soils.

## Figures and Tables

**Figure 1 plants-13-00402-f001:**
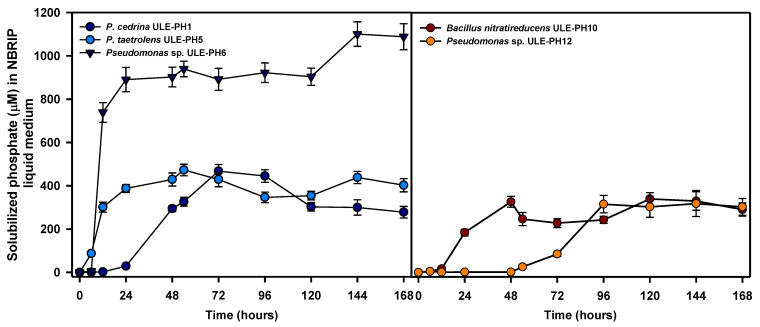
Evolution of phosphate solubilization in NBRIP liquid medium by selected bacterial isolates. The data shown (with SD) are the average of three independent experiments.

**Figure 2 plants-13-00402-f002:**
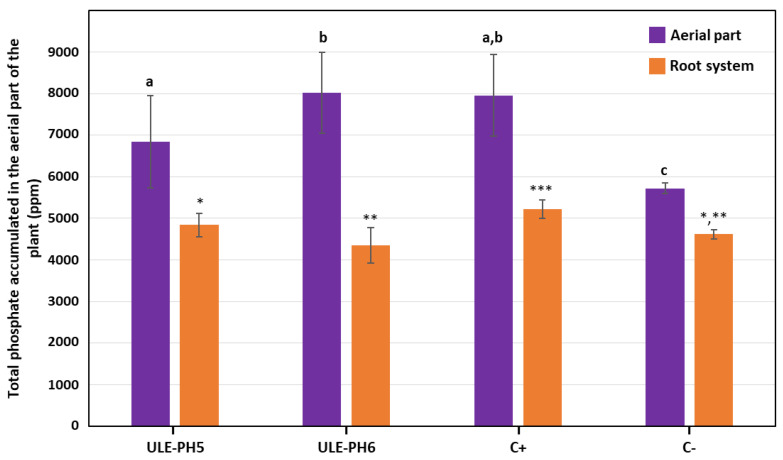
Total phosphate accumulated in the aerial part of the plant (purple bars) and the root system (orange bars) at the end of the experiment. The negative control (C−) group consisted of plants irrigated with a plant nutrient solution lacking soluble phosphate and without the application of PSBs. The positive control (C+) group consisted of plants irrigated with the same plant nutrient solution supplemented with 0.2 g/L of soluble KH_2_PO_4_. The data presented corresponds to the average of 15 plants. Bars marked with the same letter are not significantly different (*p* ≥ 0.05). Bars marked with the same number of asterisks are not significantly different (*p* ≥ 0.05).

**Figure 3 plants-13-00402-f003:**
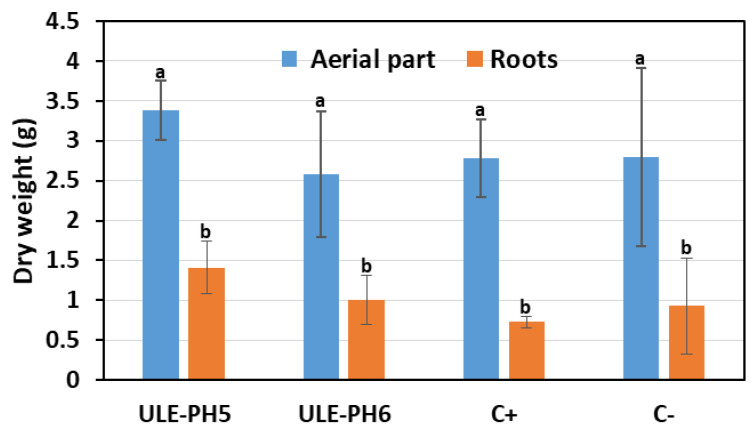
Growth data of hop plants inoculated with ULE-PH5 and ULE-PH6 strains as compared to negative control (untreated plants) (C−) and positive control (C+) plants (watered with a nutritive solution of soluble phosphate). The dry weight of the roots and aerial parts (stems, petioles, and leaves) of the plants are shown. The data indicated corresponds to the average of 15 plants. Bars marked with the same letter are not significantly different (*p* ≥ 0.05). Bars marked with the same letter are not significantly different (*p* ≥ 0.05).

**Table 1 plants-13-00402-t001:** Bacterial strains isolated from the rhizosphere of hop plants and selected based on Solubilization Index (SI) values greater than 5.0 in solid NBRIP and phytase screening media, along with their molecular identification by 16S rRNA and/or rpoD sequencing.

Isolate	16S rRNA Nucleotide Identity (%)	*rpoD*/*rpoB* ^#^ Nucleotide Identity (%)	Solubilization Index (SI) in NBRIP Medium	SI in Phytase Screening Medium (PSM)
*Pseudomonas cedrina* ULE-PH1	100.00%	99.35%	7.05 (0.11)	6.17 (0.09)
*Pseudomonas taetrolens* ULE-PH5	>99.60% *	98.81%	7.38 (0.09)	7.63 (0.15)
*Pseudomonas* sp. ULE-PH6	>99.80% *	>98.00% *	8.31 (0.10)	8.75 (0.12)
*Bacillus nitratireducens* ULE-PH10	>99.90% *	99.46%	7.42 (0.16)	5.04 (0.09)
*Pseudomonas* sp. ULE-PH12	>99.90 *	>98.00% *	5.67 (0.10)	5.06 (0.07)

^#^ *rpoD* sequences were used for discrimination between *Pseudomonas* sp. species, whereas *rpoB* sequences were used for the identification of *Bacillus* sp. species. * Several bacterial species exhibited nucleotide homology greater than the indicated value.

**Table 2 plants-13-00402-t002:** Characterization of biochemical and physiological traits of *P. taeotroles* ULE-PH5 and *Pseudomonas* sp. ULE-PH6 strains by using a standard API^®^ 20 NE test.

Test	Reaction/Enzyme	*P. taetrolens* ULE-PH5 *	*Pseudomonas* sp. ULE-PH6 *
NO_3_	Reduction of nitrate to nitrite	−	+
NO_3_	Reduction of nitrite to N_2_	−	−
TRP	Indol production (tryptophanase activity)	−	−
GLU ^#^	Glucose fermentation	−	−
ADH ^#^	Argini ne dihydrolase	+	+
URE ^#^	Urease	−	−
ESC	Esculin hydrolisis (β-glucosidase)	−	−
GEL	Gelatin hydrolisis (protease)	−	−
PNPG	β-galactosidase	−	−
GLU	Glucose assimilation	+	+
ARA	Arabinose assimilation	+	+
MNE	Mannose assimilation	+	+
MAN	Mannitol assimilation	+	+
NAG	N-acetyl glucosamine assimilation	−	+
MAL	Maltose assimilation	−	−
GNT	Gluconate assimilation	+	+
CAP	Capric acid assimilation	+	+
ADI	Adipic acid assimilation	−	−
MLT	Malic acid assimilation	+	+
CIT	Citrate assimilation	+	+
PAC	Phenylacetic acid assimilation	−	−
OX	Citochrome oxidase	+	+

* +, positive result; −, negative result. # The underlining of GLU, ADH and URE tests indicate that they were performed under anaerobic conditions by filling the domes of the microtubes with oil paraffin.

**Table 3 plants-13-00402-t003:** Miscellaneous traits and activities related to PGPR of the selected *P. taeotroles* ULE-PH5 and *Pseudomonas* sp. ULE-PH6 strains.

Miscellaneous Traits Related to PGPR	*P. taetrolens* ULE-PH5	*Pseudomonas* sp. ULE-PH6
Siderophores production	+	+
HCN production	−	+
IAA production (μg/mL)	59.29 ± 2.12	30.69 + 2.03
Zn solubilization efficiency	−	−
K solubilization efficiency (%)	5.88 ± 0.30	6.26 ± 0.18
Extracellular acidic phosphatase (mU/μg protein)	90.22 ± 17.37	98.04 ± 6.95
Extracellular alkalyne phosphatase (mU/μg protein)	49.84 ± 12.43	34.95 ± 1.27
Intracellular acidic phosphatase (mU/μg protein)	34.30 ± 2.99	92.09 ± 12.93
Intracellular alkalyne phosphatase (mU/μg protein)	48.63 ± 16.09	52.54 ± 1.23
Phytase activity (mU/mL)	40.25 ± 1.71	43.38 ± 2.75

**Table 4 plants-13-00402-t004:** GenBank Accession numbers of the different DNA sequences used for the molecular identification of PSBs.

Strain	Genbank Accession Numbers		
	16S rRNA	*rpoD*	*rpoB*
*Pseudomonas cedrina* ULE-PH1	OR946218	OR965860	
*Pseudomonas taetrolens* ULE-PH5	OR946219	OR965861	
*Pseudomonas* sp. ULE-PH6	OR946220	OR965862	
*Bacillus nitratireducens* ULE-PH10	OR946221		OR965864
*Pseudomonas* sp. ULE-PH12	OR946222	OR965863	

## Data Availability

Data is contained within the article.

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
