# Peer review of "Pseudomonas taetrolens ULE-PH5 and Pseudomonas sp. ULE-PH6 Isolated from the Hop Rhizosphere Increase Phosphate Assimilation by the Plant"

_plants, 2024, doi:10.3390/plants13030402_

Round 1
Reviewer 1 Report
Comments and Suggestions for Authors
The manuscript "Pseudomonas taetrolens ULE-PH5 and Pseudomonas sp. ULE—2 PH6 isolated from the rhizosphere of hop plants increase phos-3 phate assimilation by the plant 4" showed that the study characterized phosphate-solubilizing bacteria (PSB) from rhizosphere of hop plants.
The bacterial Pseudomonas strains found to be high significance in promoting plant growth by due to their efficient capability in solubilizing and high uptake of phosphate as shown in plate and liquid culture assays. The study has good potential in producing a biofertilizers in the future. The manuscript may not required any revisions.
Author Response
Dear Reviewer:
Thank you very much for your positive opinion of our manuscript
Reviewer 2 Report
Comments and Suggestions for Authors
Dear Author,
Please see the comments in the attached PDF.

Author Response
Dear Reviewer: thank you very much for your helpful suggestions to improve the manuscript. Below you can find our answer to the different points/questions you raised:
+ Page 1, line 22: We have mentioned now the aerial parts of the plant were P accumulation was detected.
+ Page 2, lines 78-79. The indicated modification has been made
+ Page 6, lines 202-204. Numerical data indicating the low antifungal activity of both isolates against the fungal pathogens tested are now indicated in the text
+ Page 7, lines 252-256. Details regarding Negative (C-) and Positive (C+) controls have now been included in the legend of Figure 2
+ Page 10, lanes 392-404. A more detailed explanation of the methodology for the isolation of bacteria from the rhizosphere of hop plants is now made.
Reviewer 3 Report
Comments and Suggestions for Authors
The title may be revised to "Pseudomonas taetrolens ULE-PH5 and Pseudomonas sp. ULE— 3 PH6 isolated from the hop rhizosphere increases phosphate assimilation".
The introduction needs to be revised and authors may incorporate molecular mechanisms for phosphate solubilization and mobilization.
Discussion needs to improve by citing recent research highlighting the mechanism.
Author Response
Dear Reviewer:
Thank you very much for your positive comments on our manuscript
+ As you suggest the title of the manuscript has been modified
+ The Introduction has been modified to describe in brief the main molecular mechanisms used for microorganisms to promote phosphate solubilization and mobilization and including some recent references
+ As you suggest the Discussion section has been improved by citing some recent research highlighting some of the mechanisms used by PSBs to solubilize and mobilize phosphorus